# A Model-Based System for Real-Time Articulated Hand Tracking Using a Simple Data Glove and a Depth Camera

**DOI:** 10.3390/s19214680

**Published:** 2019-10-28

**Authors:** Linjun Jiang, Hailun Xia, Caili Guo

**Affiliations:** 1Beijing Key Laboratory of Network System Architecture and Convergence, School of Information and Communication Engineering, Beijing University of Posts and Telecommunications, Beijing 100876, China; j785137381@gmail.com (L.J.); guocaili@bupt.edu.cn (C.G.); 2Beijing Laboratory of Advanced Information Networks, Beijing 100876, China

**Keywords:** articulated hand tracking, multi-model, data glove, depth camera, model-fitting, real-time

## Abstract

Tracking detailed hand motion is a fundamental research topic in the area of human-computer interaction (HCI) and has been widely studied for decades. Existing solutions with single-model inputs either require tedious calibration, are expensive or lack sufficient robustness and accuracy due to occlusions. In this study, we present a real-time system to reconstruct the exact hand motion by iteratively fitting a triangular mesh model to the absolute measurement of hand from a depth camera under the robust restriction of a simple data glove. We redefine and simplify the function of the data glove to lighten its limitations, i.e., tedious calibration, cumbersome equipment, and hampering movement and keep our system lightweight. For accurate hand tracking, we introduce a new set of degrees of freedom (DoFs), a shape adjustment term for personalizing the triangular mesh model, and an adaptive collision term to prevent self-intersection. For efficiency, we extract a strong pose-space prior to the data glove to narrow the pose searching space. We also present a simplified approach for computing tracking correspondences without the loss of accuracy to reduce computation cost. Quantitative experiments show the comparable or increased accuracy of our system over the state-of-the-art with about 40% improvement in robustness. Besides, our system runs independent of Graphic Processing Unit (GPU) and reaches 40 frames per second (FPS) at about 25% Central Processing Unit (CPU) usage.

## 1. Introduction

The articulated hand tracking has been studied for decades, due to its wide range of applications, like computer graphics, animation, human-computer interaction, rehabilitation, and robotics. Nowadays, with the boom of virtual reality (VR) and augmented reality (AR), more natural interaction with the digital world is desired to increase the sense of presence and immersion. Fully articulated hand tracking holds the potential to become a first-class input mechanism [1]. Recent works have put more attention on the task of fully articulated hand tracking, aiming to recover the detailed motion of a user’s hands in real-time. However, tracking detailed hand motion is still a challenge, facing many factors like large variations in hand shapes, small hand size, viewpoint changes, many degrees of freedom (DoFs), fast movement, self-similarity, and occlusions [2].

Judging by the input devices, we can broadly categorize existing works into the wearable-based and the camera-based. For the wearable-based works, the data gloves that can record hand pose stably and directly, are the most representative. When it comes to those camera-based researches, an approach to recover hand pose is using discriminative (appearance-based) methods [3]. These methods search the correspondence between input images and pose parameters from a large amount of data using deep-learning. Another approach is the use of generative (model-based) methods [4,5,6] that apply iterative model-fitting optimization to camera images. Due to their heavy dependence upon the proper initialization, recent works [7,8] utilize the appearance-based methods to provide initialization and re-initialization. Apart from the wearable-based and camera-based methods, some researchers [9,10] try to make improvements through multi-model input devices.

Although the methods above have made many significant contributions, there is still a long way for fully articulated hand tracking to become the first choice of the user interface. The following are several factors. For data-glove based systems, a complex, tedious, non-automatic calibration is inevitable to reduce inaccuracies, especially for applications that require high levels of accuracy [11]. Those expensive commercial gloves, i.e., 5DT Data Glove [12], Cyber Glove [13], are always unaffordable. They also add on additional hardware which may cause discomfort to the user for prolonged use. For camera-based works, discriminative methods still generalize poorly to unseen hand shapes [14,15] because existing hand data-sets have many shortcomings, e.g., low variation in hand shape, annotation accuracy, and limited in scale. Most generative methods are computationally expensive and a top-end consumer GPU is essential for heavy parallelization to achieve real-time performance. The inherent issue of visual occlusions also makes those camera-based works still lack sufficient accuracy and robustness for fine manipulation, like surgical applications. As for those multi-model researches, their systems are always burdensome because of their poor integration of various inputs. All those factors mean there is still room to make further improvement in flexibility, accuracy, robustness, efficiency, and cost.

In this paper, we present a system with the synergy of a simple data glove and a depth camera to recover articulated hand motion by iterative model-based registration progress. We aim to achieve more accurate and robust tracking results and overcome some downsides of existing works, e.g., tedious calibration or high dependency on GPU. The contributions of this paper are as follows:Design and implementation of a multi-model articulated hand tracking system that runs in real-time without GPU and improves by about 40% of robustness with comparable or increased accuracy over the state-of-the-art.The re-definition and simplification on the function of the data glove as an approximate initialization in Section 3.2.2 and a strong pose-space regularization in Section 3.3.3 that increases the robustness of our system and frees our data glove from tedious calibration and heavily hampering hand movement.A new proposal for DoFs setting that can avoid potential artificial error in the kinematic chain; see Section 3.1.The new fitting terms with a simplified approach for computing tracking correspondences that can reduce computational cost without the loss of accuracy; see Section 3.3.1.A new strategy to consider the shape adjustment of the triangular mesh hand model that includes a tailored shape integration term in Section 3.3.2 for better fitting the input images and an adoptive collision prior consistent with the shape adjustment in Section 3.3.3 to prevent self-intersection and produce plausible hand poses.

The following structure of the paper is: We survey related works in Section 2. In Section 3, we introduce the components of our system, including the hand model, the setting of the kinematic chain and DoFs, data acquirement and processing, and the objective function. In Section 4, we quantitatively and qualitatively analyze the performance of our hand tracking system and provide comparisons with the state-of-the-art. We conclude in Section 5 with a discussion of future works.

## 2. Related Works

In this section, we put our focus on the relevant introduction of the two mainstream camera-based works, i.e., appearance-based methods and model-based methods. We also briefly introduce some multi-model methods. Dipietro et al. [11] have done elaborate research for all kinds of data gloves and relevant applications. We refer the readers to [11] for a detailed review of glove-based works.

### 2.1. Appearance-Based Methods

Appearance-based methods train a classifier or a regressor to map image features to hand poses. Nearest neighbor search and decision trees [16,17] are widely used in early works. In recent years, convolutional neural network (CNN)-based discriminative methods [3,18,19,20,21,22] are state-of-the-art which estimate 3D joint positions directly from depth images. Besides, kinematics and geometric constraints are considered to avoid joint estimations violating kinematic constraints. Malik et al. [2] embedded a novel hand pose and shape layer inside CNN to produce not only 3D joint positions but also hand mesh information. For a more comprehensive analysis and investigation of the state-of-the-art along-with future challenges, we refer the readers to [15]. The biggest limitation of appearance-based methods is the training data. Existing benchmarks [17,23,24,25,26] are not perfect enough to ensure well generalize to unseen hand shapes. We refer to [14] for a detailed analysis of the drawbacks of existing data-sets. Considering this limitation, our system follows the model-based approaches that do not rely on massive data-sets.

### 2.2. Model-Based Methods

Despite the considerable advance in learning-based hand tracking, systems that employ generative models of explicit hand kinematics and surface geometry and fit these models to depth data using local optimization have produced the most compelling results [1]. The most common problems for model-based methods are a good enough initialization point, an expressive enough hand model and a discriminative object function that minimizes the error between the 3D hand model and the observed data.

#### 2.2.1. Initialization

A good enough Initialization has been proven critical to the robustness [23], which enables faster converge and better resistant to local optima. There exist many initialization methods. Some works [5,23,27] were initialized by the fingertip detection. Besides, Tagliasacchi et al. [27] and Tkach et al. [5] also detected a color wristband as a first alignment. The use of simple geometric heuristics for initialization can sometimes be impractical for those gestures which contain occlusions or difficult hand orientations. For this reason, most of the previous studies concentrated on exploiting the given image data with the train-based methods. Taylor et al. [28] generated candidate’s hand poses quickly by a retrieval forest [29]. Taylor et al. [1,30] trained a decision forest classifier on a synthetic training set to generate an initial pose estimate. Sanchez-Riera et al. [7] trained a convolutional neural network for initialization with 243,000 tuples of images. Sharp et al. [6] inferred a hierarchical distribution over hand pose with a layered discriminative model. However, initialization errors often occur due to imperfect training data-sets, mentioned in Section 2.1, which may cause tracking failure. In our system, it is more reliable and robust to provide an approximate initialization by a simple data glove.

#### 2.2.2. Hand Model

The human hand model serves as the medium of computation and the presentation of algorithm results. A detailed and accurate generative model tends to deepen the good local minima and widen their basins of convergence [1]. Many hand models have been proposed, see Figure 1.

Early works [27,31,32,33] used the capsule mode made by two basic geometric primitives: a sphere and a cylinder. Qian et al. [23] built the hand model using a number of spheres. Melax et al. [34] used a union of convex bodies for hand tracking. Sridhar et al. [35] modeled the volumetric extent of the hand as a 3D sum of an-isotropic Gaussian model. These approaches can model a broad spectrum of hand shape variations and enable fast evaluation of distances and a high degree of computational parallelism. However, they only roughly approximate hand shape even if Tkach et al. [5] proposed the use of sphere-meshes as a novel geometric representation. An alternative is a triangulated mesh model [4,6,7,8,28,30,36,38] with linear blend skinning (LBS) that is more realistic and fits image data better. But these triangulated meshes cost more computational effort and are hard to deal with the collision. There also exist some implicit templates except these explicit models. Schmidt et al. [37] voxelized each shape-primitive and computed a signed distance function for the local coordinate frame. Taylor et al. [1] constructed the hand as an articulated signed distance function that allows fast calculation of the distance to the hand surface. To explain the input data better and explicitly visualize the tracking result, our system uses an expressive triangular mesh hand model.

Apart from modeling the hand model detailed and realistic, hand model personalizing is a core ingredient in model-based methods. Joseph Tan et al. [38] quantitatively demonstrated for the first time that detailed personalized models improve the accuracy of hand tracking. In some works [7,27], only the simple uniform scaling of the model was considered. Makris et al. [33] investigated the calibration of a cylinder model through particle swarm optimization. Ballan et al. [4,8] reconstructed a personalized template mesh offline with a multi-view stereo method for more detail model calibration. Taylor et al. [30] presented a fast, practical method to acquire detailed hand models from as few as 15 frames of depth data. Moreover, their work was extended in [36], which simplifies hand shape variation by linear shape-spaces. Joseph Tan et al. [38] went further and presented a fast, practical method for personalizing a hand shape basis to an individual user’s detailed hand shape using only a small set of depth images. Similarly, Remelli et al. [39] presented a robust algorithm for personalizing a sphere-mesh tracking model to a user from a collection of depth measurements. However, those methods suffer a major drawback: the template must be created during a controlled calibration stage, where the hand is scanned in several static poses (i.e., offline). Tkach et al. [40] yielded a fully automatic, real-time hand tracking system that jointly estimates pose and shape for their sphere-mesh hand model. It remains unsolved for the triangular mesh model to adjust its shape during tracking. Our system tries to adapt the conclusion in [40] for the shape adjustment of our triangular mesh model to fit the shape of input images.

#### 2.2.3. Objective Function

The objective function measures the discrepancy between the hand model and input depth, as well as the validity of the hand model [23]. In general, the objective function is made by fitting terms and prior terms.

Fitting terms measure how well the hand parameters explain the input frames. Oikonomidis et al. [31] and Sharp et al. [6] formed the fitting terms as the discrepancy between the observed images and the rendered images from a given hand pose hypothesis. Then solutions were searched by a slow-converge stochastic optimizer like PSO. Most works [5,7,23,27,28,30,32,36] modeled the fitting terms as the least-squares error between the effectors and their target positions, and solved it by gradient-based approaches, such as Gauss–Newton and Levenberg–Marquardt. However, finding corresponding points is difficult and time-consuming, especially for triangular hand models. Taylor et al. [28,30,36,38] subdivided the mesh to produce a smooth surface function for evaluating both pose and corresponding points. Down-sampling the point cloud randomly is also a good way to reduce computation effort [23,28]. Except for the 3D fitting term, a 2D registration in most works [4,5,23,27,28,39,40] is also important, which pushes the hand model to lie within the visual sensor hull. Our system also adopts the 3D and 2D fitting terms but deals with the time-consuming problem of computing corresponding points to reduce the computational cost and enable our system independent of GPU.

Prior terms regularize the solution to produce realistic hand poses. Every model-based works adopt a joint limitation term extracted from a database to constrain the posture parameters within plausible value ranges. Self-intersection is a big problem. Oikonomidis et al. [31] penalized abduction–adduction angles of adjacent fingers. Some works [5,27,39,40] restricted the distance between cylinders to solve this problem. Qian et al. [23] limited the distance of spheres in neighboring fingers. For triangular hand models, it becomes harder. In [4], a repulsion term was computed in the form of a 3D-3D correspondence that pushes the vertex back. Taylor et al. [28] defined a set of spheres that approximate the volume of the fingers to simplify this question. However, these collision terms did not consider the shape adjustment during tracking. Our system gets this point and introduces an adaptive collision term consistent with shape adjustment. Besides, the pose-space prior in [5,27,28,38], obtained by performing dimension reduction on the training data, provides implicitly constrains to the recovered hand postures. In general, the pose-space prior covers a large pose space and can not constrain the hand pose tightly. Our system integrates a data glove in it to produce a stronger implicit restriction. There are also lots of prior terms, including temporal priors to prevent the tracked hand to jitter [5,27,28], ARAP regularization to penalized large shape deformations [30,36], fingertips prior term to guarantee each detected fingertip should have a model fingertip nearby [28].

### 2.3. Multi Model Systems

For multi-model systems, the core idea is that different input models each have their limitations, but may complement each other. For example, wearable-based systems can fill in the data gap that occurs with vision-based systems during camera occlusions, and the vision-based device provides an absolute measurement of hand state [9]. Arkenbout et al. [9] integrated the hand pose of 5DT date glove and the Nimble VR system through a Kalman filter and shows substantial improvement in accuracy. Ponraj et al. [10] increased the accuracy of fingertips tracking in occluded cases by combining the leap motion control with a Sensorized Glove. Tannous et al. [41] proposed a fusion scheme between inertial and visual motion capture sensors to improve the estimation accuracy of knee joint angles. Sun et al. [42] reached a higher gesture recognition rate using the Kinect and Electromyogram signals. Pacchierotti et al. [43] placed a novel wearable cutaneous device on the proximal phalanx to improve the tracking of the fingertips on commercially-available tracking systems, such as the Leap Motion controller or the Kinect sensor. These methods show improvement through multi-model inputs. These multi-model systems show improvement in accuracy and robustness, but they retain the cumbersome setting of wearable devices, e.g., calibration and uncomfortable hardware, which makes their system more burdensome. Our system can reduce the effect of additional hardware by re-defining and simplifying the function of the data glove to make our system lightweight.

## 3. Method

The overview of our system can be found in Figure 2. In this section, we will introduce our system in detail. We first present the hand model and the setting of the kinematic chain before tracking in Section 3.1. Then, we introduce step 1 during tracking, i.e., how we acquire and process the input data from the depth camera and the simple data glove, in Section 3.2. Finally, we describe in detail the step 2 during tracking, i.e., the objective function constructing, in Section 3.3. The iterative optimization step by Levenberg–Marquardt approach is not included in this Section because it is a very common gradient-based solution.

### 3.1. Hand Model

We use the publicly available MANO hand model [44] for pose and shape tracking, see Figure 3a,b. There are several reasons: (1) it is learned from around 1000 high-resolution 3D scans of hands of 31 subjects in a wide variety of hand poses. (2) As an articulated triangular model, it is more expressive than those hand models [1,2,4,7,8,14,15] made by basic geometric primitives. It also reduces the artifacts of LBS, i.e., mesh collapse around joints. (3) Romero et al. [44] provide not only a set of shape offset vectors to generate different hand shapes but also the sparse linear joints locations regressor to generate corresponding hand skeleton, which is important for shape tracking.

The general formulation $M\left(\beta ,\theta \right)$ of MANO, taken from the original paper [44] for completeness, is as follows:(1)$$M\left(\beta ,\theta \right)=W\left(T\left(\beta ,\theta \right),J\left(\beta \right),\theta ,W\right)$$
where β and θ control the shape and pose respectively, *W* is a linear blend skinning function applied to a template hand triangulated mesh *T*, the hand template *T* is obtained by deforming a mean mesh through β and θ, *J* is a sparse linear joints locations regressor learned from mesh vertices, and W is the blend weights. For more details about MANO, please refer to [44].

Given the MANO hand joints, we build our kinematic chain, Figure 3c.

The kinematic chain is not fixed because the shape parameters θ of MANO will affect the locations of the joints. Each joint $J_{k}$ of the kinematic chain, except the wrist joint, is defined with its previous joints $J_{k}^{parent}$. Moreover, each joint $J_{k}$ is associated with an orthogonal frame ${\bar{T}}_{k}$ according to which local transformations are specified. We automatically set the local coordinate systems ${\bar{T}}_{k}$ according to the relationship and structure of joints in the mean shape of the hand (β is set to 0). For the sake of simplicity, these preset local coordinate systems ${\bar{T}}_{k}$ do not change with the location change of the joints, considering the relatively small effect of β on hand structure. Incorrectly specified kinematic frames can be highly detrimental to tracking quality [5]. Relaxing the restrictions on DoFs helps reduce the impact of incorrectly setting frames, see Figure 4b.

So each joint $J_{k}$ in our kinematic chain has three DoFs, which results in 51 DoFs in total with the six DoFs global transformation. However, more DoFs mean larger search space, which is dealt with in Section 3.3.

### 3.2. Data Acquisition and Processing

#### 3.2.1. Camera Data

We get the input of depth images from Intel RealSense SR300 (Intel, Santa Clara, CA, USA), a consumer short-range RGBD camera. From the raw depth images, we obtain the segment of hand by performing classification with standard random decision forests [25]. A 2D silhouette image $S_{s}$ is extracted directly from the segment. Besides, we extract also a 3D point cloud $X_{s}$ that contains about 200 pixels by performing a stochastic sampling, which has been proved by [28] to enable CPU optimization without significant loss of precision.

#### 3.2.2. Glove Data

We measure the approximate pose of the hand, using a simple and cheap prototype data glove provided by our cooperation company [45], Figure 5a,b. The glove is affixed with 11 inertial measurement units (IMUs) according to the anatomical structure of the hand. Figure 5c shows the location, initial coordinate systems, and orientation of those IMUs.

The output of our glove is a 100 Hz stream of the orientation of IMUs in the form of unit quaternions:(2)$$Q_{i}=\left[q_{i}^{x},q_{i}^{y},q_{i}^{z},q_{i}^{w}\right],i\in \left\{1\ldots 11\right\},$$
where *i* is the index of IMUs in Figure 5c. $Q_{11}$ records the global rotation information of the hand. $Q_{1}$ and $Q_{2}$ represent the movements of the distal and proximal phalange of the thumb, respectively. The rest quaternions represent the movements of the medial and proximal phalanges of other fingers. Theoretically, all IMUs’ orientation is set to be the same in the initial pose, see Figure 5c and Figure 6a, which means initially:(3)$$Q_{i}^{initial}=Q_{j}^{initial}\;\forall i,j\in \left\{1,2,\ldots ,11\right\}.$$

Based on this premise, we can easily measure the angular values of joints, using the relative rotation between IMUs:(4)$$Q_{i,j}=Q_{i}*Q_{j}^{-1},$$
where *i* represents the index of IMUs, *j* represents the index of the parent IMU of the *i*th IMU, $Q_{i,j}$ is also a unit quaternion that contains the rotation information of the joint between the two phalanges where the *i*th and *j*th IMU locate in. See Figure 6 for example.

Then we convert the quaternion $Q_{i,j}$ to the Euler angle in XYZ rotation order and map to the 51 DoFs pose parameters mentioned in Section 3.1 to provide an initial hand pose:(5)$$\begin{array}{cc} \theta_{ij}^{x} & =tan^{-1}\left(\frac{2\left(p_{ij}^{w}p_{ij}^{x}+p_{ij}^{y}p_{ij}^{z}\right)}{1-2\left({p_{ij}^{x}}^{2}+{p_{ij}^{y}}^{2}\right)}\right)\\ \theta_{ij}^{y} & =sin^{-1}\left(2\left(p_{ij}^{w}p_{ij}^{y}-p_{ij}^{x}p_{ij}^{z}\right)\right)\\ \theta_{ij}^{z} & =tan^{-1}\left(\frac{2\left(p_{ij}^{w}p_{ij}^{z}+p_{ij}^{x}p_{ij}^{y}\right)}{1-2\left({p_{ij}^{y}}^{2}+{p_{ij}^{z}}^{2}\right)}\right). \end{array}$$

The movements of the metacarpal phalanx of the thumb and the distal phalanx of other fingers are not considered independently but calculated according to the proximal phalanx of the thumb and the medial phalanx of other fingers respectively.

Our simple data glove only provides approximate poses, see Figure 7, because the fixed positions of IMUs may not fit everyone, and shifting of IMUs may appear when incorrect wearing the glove or making various gestures.

We redefine and simplify the function of the data glove to be a robust initialization, re-initialization, and a strong prior. So we think the approximate pose from our simple data glove is acceptable and do not design a complex calibration progress for it to keep our system light-weighted and easy to use.

### 3.3. Objective Function

Given the 2D silhouette image $S_{s}$, 3D point cloud $X_{s}$, and initialization form data glove, we aim to find the pose θ and shape β parameters that make our hand model a good explanation of the absolute measurement from the camera accurately and efficiently. We formulate this goal as a minimum on the following objective function with references to recent works:(6)$$\min_{\theta ,\beta }\underset{Fitting\;terms}{\underset{︸}{E_{3D}+E_{2D}}}+E_{shape}+\underset{Prior\;terms}{\underset{︸}{E_{posespace}+E_{temporal}+E_{collision}+E_{bounds}}},$$
where fitting terms determine the pose θ and shape β parameters in each frame, $E_{shape}$ elegantly integrates shape information β from frames in different poses, those prior terms regularize the solution to ensure the recovered pose is plausible. We will put our focus on the novelties to meet our premise on efficiency, accuracy, and robustness, and give only a brief introduction on the unchanged terms.

#### 3.3.1. Fitting terms

3D Registration. The $E_{3D}$ registers the hand model M to the point cloud $X_{s}$. In recent works, it is always formulated in the spirit of ICP as:(7)$$E_{3D}=\omega_{3D}\sum_{x\in X_{s}}{∥x-\Pi_{M}\left(x\right)∥}_{2}^{2}$$
where *x* represents a 3D point of the point cloud $X_{s}$, $\Pi_{M}\left(x\right)$ is the corresponding point of *x* on the hand model M.

Finding the corresponding points is the most critical, time-consuming, and challenging step, especially for triangular mesh models. In order to improve efficiency and reduce the computational cost, we simplify this process and re-formulate the 3D registration without loss of accuracy as:(8)$$E_{3D}=\omega_{3D}\sum_{x\in X_{s}}{∥x-\Pi_{\hat{V}}\left(x,\theta ,\beta \right)∥}_{2}^{2}$$
where the difference is that we directly search the corresponding point $\Pi_{\hat{V}}\left(x,\theta ,\beta \right)$ among the visible vertices $\hat{V}$ (about half of the original 778 vertices) for each 3D point $x\in X_{s}$. The reasons behind are two-fold:The MANO hand model is more detailed than the hand models in [4,6,7,8,28,30,36,38] even though it is made by only 778 vertices and 1554 triangular faces. We also try to produce a more detailed model by the Loop subdivision [36], see Figure 8. The result shows no significant improvement, which enables the vertex v∈V account for these corresponding points around it.We conducted another way to find more detailed corresponding points on the hand model M using loop subdivision and compare the performance with our simplified approach, see Figure 9. The result shows that our approach will not decrease accuracy.

With the down-sampled point cloud $X_{s}$, this approach dramatically reduces the amount of calculation and allows us to search all the corresponding points for the point cloud $X_{s}$ in less than 1 ms on CPU with a single thread.

2D Registration. The $E_{2D}$ provides the supplementary registration that the point cloud alignment does not take into account. Ganapathi et al. [46] show that the depth map provides evidence not only for the existence of a model surface in the form of point cloud $X_{s}$ but also for the non-existence of surface between the point cloud $X_{s}$ and the camera. Thus, the $E_{2D}$, also called “free space” constraint, is non-trivial in registration. We adapt the formula mentioned in [27] for this “free space” constraint as:(9)$$E_{2D}=\omega_{2D}\sum_{p\in S_{r}\left(\theta ,\beta \right)}{∥p-\Pi_{S_{s}}\left(p\right)∥}_{2}^{2},$$
where *p* is a 2D point of the projection $S_{r}\left(\theta ,\beta \right)$ of the visible vertices $\hat{V}$ with pose θ and shape β, which means we need not rasterizing the model and significantly decrease the run-time complexity. $\Pi_{S_{s}}$ and $\Pi_{S_{s}}\left(p\right)$ remain unchanged and denote the image-space distance transform [46] and the closest 2D point in the input 2D silhouette $S_{s}$, respectively.

#### 3.3.2. Shape Integration

To explain the full input image well, we should take shape β into account as well, which has been shown to have a considerable impact on the accuracy [5]. Shape information β is so weakly constrained in any given frame that sufficient information must be gathered from different frames capturing different hand poses [40]. We refer to [40] to integrate shape information β from frames in different poses elegantly. Tkach et al. [40] do similar work on their 112 manually designed explicit shape parameters of the sphere-mesh model, i.e., the length of fingers and the radius of circles. However, our shape parameters β provided by [44] are the top 10 shape principal components analysis (PCA) components that control the shape of the hand model implicitly. In order to adapt the method in [40] to our triangular hand model, we make the following feasibility analysis:

We imitate [40] to abstract the hand shape/pose estimation problem from a single frame into one of a simpler 2D stick-figure, Figure 10.

The co-variance of shape parameters is derived from the Hessian Matrix of the registration energies. The co-variance represents the confidence or uncertainty of shape parameters. Figure 10 shows a similar conclusion in [40] that the co-variance of the implicit shape parameters is also conditional on the pose of the current frame, and the co-variance decreases with a bent finger. Besides, the implicit shape parameters can produce better convergence results even when the blend angle is small. This analysis means that we can easily adapt the joint cumulative regression scheme in [40] without changing the form:(10)$$E_{\mathrm{shape}}=\omega_{shape}{∥{\hat{\Sigma }}_{n-1}^{-1/2}\left(\beta_{n}-\hat{\beta_{n-1}}\right)∥}_{2}^{2},$$
and update the shape parameters $\beta_{n}$ and the cumulative co-variance ${\hat{\Sigma }}_{n}$ in the *n*th frame using the kalman-like style mentioned in [40]:(11)$$\begin{array}{cc} {\hat{\beta }}_{n} & =\Sigma_{n}^{*}{\left({\hat{\Sigma }}_{n-1}+\Sigma_{n}^{*}\right)}^{-1}{\hat{\beta }}_{n-1}+{\hat{\Sigma }}_{n-1}{\left({\hat{\Sigma }}_{n-1}+\Sigma_{n}^{*}\right)}^{-1}\beta_{n}^{*}\\ {\hat{\Sigma }}_{n} & ={\left({\hat{\Sigma }}_{n-1}^{-1}+{\Sigma_{n}^{*}}^{-1}\right)}^{-1}. \end{array}$$

For more details about the kalman-like shape integration, we refer the readers to [40].

#### 3.3.3. Prior Terms

Pose Space prior. The $E_{posespace}$ is a data-driven prior that limits the pose space (except the six global components of θ) within a reasonable range. We extend the pose space prior in [27] with the restriction of data glove to narrow the pose searching space.

We first construct the standard pose space prior in [27] on the publicly available database of [44] by PCA, see Figure 11a, which results in a 45×45 matrix V of eigen-vectors and a set of 45 eigenvalues $\lambda =\left(\lambda_{1},\ldots ,\lambda_{45}\right)$. Taking the top *N* pose components, we have:(12)$$\begin{array}{cc} \; & \tilde{\theta }=C^{T}\left(\theta -\mu \right)\\ \; & \tilde{\theta }∼N\left(0,\Sigma \right), \end{array}$$
where μ is the 45-dimensional mean pose, Σ is a diagonal matrix that contains the variance of the PCA basis, *C* is a 45×N matrix made by the top *N* eigen-vectors in *V* corresponding to the *N* largest eigenvalues. Tagliasacchi et al. [27] hold that the estimated pose should lie in this data-driven space to take on reasonable poses.

Then we take the input of data glove $\theta_{glove}$ into account. We think that we should also search the exact pose around $\theta_{glove}$. Thus, we project the input glove data $\theta_{glove}$ recorded when we recover the hand pose on the database of [44] into ${\tilde{\theta }}_{glove}$ with the same 45×N matrix *C*, see Figure 11b. Moreover, we build a multivariate Gaussian model for the difference between the glove data ${\tilde{\theta }}_{glove}$ and corresponding ground truth $\tilde{\theta }$ in the low-dimensional subspace, see Figure 11c:(13)$$\begin{array}{cc} \; & {\tilde{\theta }}_{diff}={\tilde{\theta }}_{glove}-\tilde{\theta }\\ \; & {\tilde{\theta }}_{diff}∼N\left(\mu_{diff},\Sigma_{diff}^{\prime }\right)\\ \; & \Sigma_{diff}^{\prime }=\Sigma_{diff}+\Sigma_{noise}, \end{array}$$
where $\mu_{diff}$ is the *N*-dimensional mean difference, the $\Sigma_{diff}$ is co-variance matrix and $\Sigma_{noise}$ represents the noise of data glove.

So given a specific input of data glove $\theta_{glove}^{i}$, we search the ground-truth pose in the distribution of both N(0,Σ) and $N({\tilde{\theta }}_{glove}^{i}-\mu_{diff},\Sigma_{diff}^{\prime })$. We thus merge the two distribution, see Figure 12:(14)$$N\left(\mu_{merge},\Sigma_{merge}\right)=N\left(0,\Sigma \right)N\left({\tilde{\theta }}_{glove}^{i}-\mu_{diff},\Sigma_{diff}^{\prime }\right).$$

We rewrite the PCA prior as:(15)$$E_{posespace}=\omega_{posespace}{\left(C^{T}\theta -\mu_{merge}\right)}^{T}\Sigma_{merge}^{-1}\left(C^{T}\theta -\mu_{merge}\right)$$

After Cholesky decomposition on $\Sigma_{merge}^{-1}=LL^{T}$, we convert the PCA prior to a squared form:(16)$$E_{posespace}=\omega_{posespace}{∥L(C^{T}\theta -\mu_{merge})∥}_{2}^{2}$$

Temporal prior. In order to mitigate jitter of hand on time series, we adopt a very efficient and effective prior from [27]. We build a set K that contains 50 randomly selected vertices from the hand model, and penalize their velocity and acceleration the same as [27]:(17)$$E_{temporal}=\omega_{temporal}\left(\sum_{v_{i}\in K}{∥v_{i}-v_{i-1}∥}_{2}^{2}+\sum_{v_{i}\in K}{∥v_{i}-2v_{i-1}+v_{i-2}∥}_{2}^{2}\right).$$

Collision prior. The $E_{collision}$ is used to avoid self-intersections of fingers. Unlike the hand model made by simple geometry, the MANO hand model is a triangular mesh model. It is a difficult problem to judge whether self-crossing occurs with little cost. To solve this problem, we follow the ideas in [28] and approximate the volume of the fingers in the MANO hand model with a set of spheres, see the right picture in Figure 13.

In order to be consistent with the pose and shape deformation of the hand model, we let those spheres have radius $r_{s}\left(\theta ,\beta \right)$ and locations $c_{s}\left(\theta ,\beta \right)$ specified by the vertices of hand model with pose θ and shape β. For each phalanx, we automatically set four spheres, one is root and the other three from interpolation:(18)$$\left\{\begin{array}{cc} c^{root} & =J\left(\theta ,\beta \right)\\ r^{root} & =min\left\{d_{i,J\left(\theta ,\beta \right)}\right\}\;i\in V_{J} \end{array}\right.\;\left\{\begin{array}{cc} c_{i}^{interpolation} & =J\left(\theta ,\beta \right)+\frac{i*\vec{JJ_{child}}}{N+1}\\ r_{i}^{interpolation} & =r_{child}^{root}+\frac{N+1-i}{N+1}\left(r^{root}-r_{child}^{root}\right) \end{array}\right.,$$
where for the root spheres, *J* is the regressor of joints, $J\left(\theta ,\beta \right)$ is the location of one joint, $V_{J}$ is the top 20 vertices that have the greatest impact on the regression of this joint in the regressor *J*, $d_{i,J\left(\theta ,\beta \right)}$ represents the Euclidean distance between the *i*th vertex in $V_{J}$ and the joint $J\left(\theta ,\beta \right)$; for the interpolated spheres, *N* represents the number of spheres you want to interpolate, $i\in \left\{1,\ldots ,N\right\}$ is the index of the interpolated sphere, $J_{child}$ is position of the child joint of *J* according the kinematic tree. See Figure 13 for details.

Given the set of spheres, we penalize self-intersection with:(19)$$E_{collision}=\omega_{collision}\sum_{i,j}X\left(i,j\right){∥p_{i}-p_{j}∥}_{2}^{2}$$
where $p_{i}$ and $p_{j}$ are two point on the *i*th sphere and *j*th sphere and play the role as original point and target point respectively, $X\left(i,j\right)$ indicates whether collision between *i*th sphere and *j*th sphere happened:(20)$$X\left(i,j\right)=\left\{\begin{array}{cc} 0 & otherwise\\ 1 & {\left(c_{i},c_{j}\right)}^{2}<{\left(r_{i}+r_{j}\right)}^{2} \end{array}\right.$$

Joints bounds prior. To prevent the hand from reaching an impossible posture by over-bending the joints, we limit the angles of the hand model and adopt the same function in [27]:(21)$$E_{bounds}=\omega_{bounds}\sum_{\theta_{i}\in \theta }\left(\underline{X}\left(i\right){\left(\theta_{i}-\underline{\theta_{i}}\right)}^{2}+\bar{X}\left(i\right){\left((\theta_{i}-\bar{\theta_{i}}\right)}^{2}\right),$$
where each hand joint is associated with limitations $\left[\underline{\theta_{i}},\bar{\theta_{i}}\right]$. Because of our different settings of DoFs, we extract the limitations for each DOF from the detailed hand database in [44]. The extracted limitations can be see as Table 1. $\underline{X}\left(i\right)$ and $\bar{X}\left(i\right)$ are indicator functions:(22)$$\begin{array}{cc} \underline{X}\left(i\right) & =\left\{\begin{array}{cc} 0 & otherwise\\ 1 & \theta_{i}<\underline{\theta_{i}} \end{array}\right.\\ \bar{X}\left(i\right) & =\left\{\begin{array}{cc} 0 & otherwise\\ 1 & \theta_{i}>\bar{\theta_{i}} \end{array}\right. \end{array}$$

## 4. Experiments and Discussion

In this section, we evaluate our system in many aspects, e.g., the robustness to noise and occlusion, the accuracy on various poses and shapes, and the improvement over the-state-of-art. The quantitatively experiments are conducted on the synthetic data-set, Handy (Teaser [5] and GuessWho [40]) data-set, and the NYU data-set [25] for self-evaluation, the comparison with model-based techniques [5,39,40], and the comparison with appearance-based works [2,20,21,22,47,48,49] , respectively. We also qualitatively show the real-time performance and comparison with [2,5,39,40].

### 4.1. Data-Sets

Synthetic data-set The Synthetic data-set was generated based on a sequence of real hand motion. Firstly, we tracked a sequence of real hand motion and recorded the recovered shape and pose parameters with the inputs of data glove synchronously. Secondly, we chose five different hand shapes from the data-set of [44]. Then we applied those pose parameters to our hand model along with the five different shape parameters and produced five sets of synthetic image sequences. Each sequence contains 1129 synthetic depth images. Moreover, we applied Gaussian noise to shape 1 with a standard deviation ranging from 0 to 12 mm. The influence of noise can be seen in Figure 14.

Handy/Teaser and Handy/GuessWho data-set. the Handy data-set was created by [5] for the evaluation of high-precision generative tracking algorithms. It contains the full range of hand motion that has been studied in previous researches. The recording device was an Intel RealSense SR300, which was the same camera used in our system. The Teaser sequence [5] among Handy contains 2800 images of one subject with various range of hand motion. The GuessWho sequence [40] among Handy contains hand movement sequences of 12 different users. For each subject, there are about 1000 to 3000 images in Handy/GuessWho data-set.

NYU data-set. The NYU data-set, introduced in [25], records a great amount of high noise real data with quite an accurate annotation. It covers a good range of complex hand poses and a wide range of viewpoints. Because the NYU data-set is full of noise and miss pixels, it is a very challenging data-set. We only use the test set of the NYU data-set that contains 8225 images from two different actors.

### 4.2. Quantitative Comparison Metrics

The metrics are chosen to explain the difference between the recovered hand motion with the original hand motion. Different data-sets offer the original hand motion in different ways. Thus the following metrics vary with each data-set.

Metrics for the synthetic data-set. The synthetic data-set is generated from our hand model. We record the original full 3D hand motion in terms of the location of vertices and joints of the hand model. So, we compute the mean errors of the vertices and joints between the recovered hand model and the ground truth to show the difference:(23)$$\begin{array}{cc} V_{error} & =\frac{\sum_{i=1}^{N_{v}}{∥V_{recovered}^{i}-V_{groundtruth}^{i}∥}_{2}}{N_{v}}\\ J_{error} & =\frac{\sum_{i=1}^{N_{j}}{∥J_{recovered}^{i}-J_{groundtruth}^{i}∥}_{2}}{N_{j}}, \end{array}$$
where $N_{v}=778$ is the number of vertices of our hand model, $N_{j}=16$ is the number of joints of our hand model.

Metrics for the Handy data-set. The Handy data-set records depth and color images for the original hand motion. Those model-based techniques [5,39,40] also provide their tracking results in the form of rendered depth images. To compare with those methods, we choose the algorithm agnostic metrics $E_{3D}$ and $E_{2D}$ proposed by [5] for evaluating the discrepancy between the input depth image and the rendered depth image. The $E_{3D}$ can be formulated as follows:(24)$$E_{3D}=\frac{\sum_{i=1}^{N_{3D}}{∥p^{i}-p_{closest}^{i}∥}_{2}}{N_{3D}},$$
where the $p^{i}$ is the *i*th point in the 3D point cloud from the input depth image, the $p_{closest}^{i}$ is the closest correspondence point of $p^{i}$ in the 3D point cloud from the rendered depth image, the $N_{3D}$ is the total number of points in the 3D point cloud from the input depth image.

The $E_{2D}$ is evaluated using follow equation:(25)$$E_{2D}=\frac{\sum_{i=1}^{N_{2D}}{∥p_{render}^{i}-p_{closest}^{i}∥}_{2}}{N_{outside}}$$
where the $p_{render}^{i}$ is the *i*th point in the 2D hand image rendered from the hand model, $p_{closest}^{i}$ is the 2D closest point of $p_{render}^{i}$ in the silhouette of the input image. $p_{closest}^{i}=p_{render}^{i}$ if $p_{render}^{i}$ lies inside the silhouette of the input image, the $N_{2D}$ is the number of 2D rendered points, the $N_{outside}$ counts only the number of 2D rendered points outside the silhouette of the input image.

Metrics for the NYU data-set. The NYU data-set provides a quite accurate annotation of joints as the ground truth of the original hand motion. Those appearance-based works [2,20,21,22,47,48,49] also predict the hand joints as the tracking results. For the comparison with [2,20,21,22,47,48,49] on the NYU data-set, we choose the widely used metric among appearance-based methods, i.e., the mean error of the 3D joint locations:(26)$$E_{mean}=\frac{\sum_{i=1}^{N}{∥J^{i}-J_{correspond}^{i}∥}_{2}}{N}$$
where the $J^{i}$ is the *i*th ground truth joint, the $J_{correspond}^{i}$ is the corresponding joint of $J^{i}$, and *N* is the total number of ground truth joints.

### 4.3. Quantitative Experiments

Synthetic data-set. We make an overall evaluation of our system on various poses, hand shapes, and noise. Figure 15 shows the performance of our system on the five hand shapes and various hand poses. It is acceptable that the $V_{error}$ and $J_{error}$ fluctuate within 3 mm for different hand shapes. Over 95 percent of the $V_{error}$ and $J_{error}$ are within 5 mm, and almost 100 percent of the the $V_{error}$ and $J_{error}$ are within 10 mm. Few outliers appear but stay in 16 mm. Besides, the robustness of our system to different hand shapes and poses is illustrated by the steep curves of the $V_{error}$ and $J_{error}$, and the narrow interquartile range(IQR) in the box plots of the $V_{error}$ and $J_{error}$. Figure 16 illustrates the robustness of our system on different noise. Our system is robust as the standard deviation of noise gradually increases from 0 mm to 8 mm. The $V_{error}$ and $J_{error}$ slightly increase but are still within 10 mm. With more severe noise, the details of the point cloud from depth image disappear, Figure 14, and there is no wonder that the performance declines sharply. However, we can still give an approximate hand pose from the data glove even though the noise makes the point cloud a mess, see Figure 14.

Handy data-set. We compare our system with state-of-the-art generative methods [5,39,40] in the aspects of accuracy and robustness on various hand poses and shapes. We acquire the tracking result of [5] on the Handy/Teaser sequence though their released code on the Internet. The authors in [39,40] provide their tracking results on the Handy/GuessWho sequence.

Figure 17 shows the comparison between our system with [5] on the Handy/Teaser sequence. It can be seen in Figure 17 that our system achieves a visually noticeable improvement on the $E_{3D}$ and a comparable result on the $E_{2D}$. The numerical improvements can be found in Table 2. The improvement of accuracy on the $E_{3D}$ is attributed to our hand model with the relaxed DoFs. Our triangular hand model deformed by LBS can express the surface geometry of the human hand better than the hand model made by sphere mesh. The relaxed DoFs introduced in Section 3.1 provide more possibilities to register the point cloud well. The comparable result on the $E_{2D}$ indicates that our system successfully tracks the hand shape in the Handy/Teaser sequence. There is an improvement of robustness on the $E_{3D}$ and $E_{2D}$ to our data glove. We owe it to our data glove. Our data glove can provide stable initialization and restriction, which reduces tracking failures, i.e., the outliers in the box plot of Figure 17. The *p*-values of significance tests for the results in the Handy/Teaser in Table 3 validate that our improvements are statistically significant.

Figure 18 gives the comparison with [39,40] on all subjects in the Handy/GuessWho sequence. The numerical improvements can be found in Table 4. We can draw a similar conclusion in terms of $E_{3D}$. When comes to the $E_{2D}$, our system shows no significant improvement, which indicates that our system does not track the hand shape well. We ascribe this to our implicit shape parameters from PCA which may overly constrain the hand shape and are hard to perform as well as the explicit 114 DoFs shape parameters on the detail of hand shape. For the comparison of each subject in the Handy/GuessWho sequence can be found in Figure 19 and Figure 20. The *p*-values of significance tests for the results in the Handy/GuessWho in Table 3 validate that our improvements of the $E_{3D}$ are statistically significant, and the performance of our system and [40] on the $E_{2D}$ is comparable with no statistical difference.

NYU data-set. We compare our system with state-of-the-art appearance-based methods [2,20,21,22,47,48,49] on the NYU data-set, see Figure 21. The results of [2,20,21,22,47,48,49] are publicly available on the Internet. Since the annotation scheme of NYU is different from ours, we manually choose a subset of 15 joints from the NYU annotation for the comparisons. Figure 22 shows the difference between the chosen joints and the joints of our hand model. Figure 21 shows the comparisons between our system with those appearance-based methods [2,20,21,22,47,48,49]. The numerical improvements can be found in Table 5. Our system achieves a comparable accuracy over those appearance-based methods. We expect better results for consistent annotation schemes. Besides, our system significantly improves about 40% on robustness. The *p*-values of significance tests for the results of the NYU dataset in Table 6 validate that the our improvements are statistically significant. Malik et al. [2] evaluated not only the hand pose but also the hand shape by deep-learning methods. We also conducted a comparison with [2] on the $E_{3D}$ and $E_{2D}$ metrics, see Figure 23. We outperform than [2] both on $E_{3D}$ and $E_{2D}$, which indicates the superiority of our model-based system on recovering the hand shape.

### 4.4. Qualitative Experiments

We first qualitatively show our real-time performance in Figure 24. We can see that our system acts well on various hand poses and the rendered depth image from the recovered hand model looks almost the same as the input depth image of the hand. We also present the robustness of our system with occlusion in Figure 25. We do not conduct extra segments for those small items, which means those small items will affect the completeness of the point cloud and confuse the tracking progress. Figure 25 shows that our system stays robust to those influences thanks to the strong prior to our data glove. Then we exhibit the qualitative comparisons with Tkach et al. [5] on the Handy/Teaser, Remelli et al. [39] and Tkach et al. [40] on the Handy/GuessWho, and Malik et al. [2] on the NYU data-set, in Figure 26, Figure 27 and Figure 28. Those qualitative comparisons also show comparable performance with state-of-the-art.

### 4.5. System Efficiency

The machine we use is a laptop with a 6-core Intel Core i7 2.2 GHz CPU, 8 G RAM, and one GPU of NVIDIA GTX1050Ti. Our system does not use GPU and occupies about 25% of CPU and about 400 MB RAM, which leaves enough CPU, RAM, and GPU for other applications. Our system can reach real-time performance at around 40 FPS. Data Acquisition and Preprocessing take less than 500 us. We solve the model-fitting of our system iteratively with a Levenberg-Marquardt approach. In general, five iterations are enough. Each iteration costs within 5 ms. In each iteration, tracking correspondences searching costs less than 1 ms. Comparing with those generative methods [5,39,40] which rely on heavy parallelization and high-end GPU hardware, it is reasonable to believe that our system is more efficient.

## 5. Conclusions and Future Works

In this paper, we propose a model-based system for real-time articulated hand tracking with the synergy of a simple data glove and a depth camera. We redefine and simplify the data glove as a strong priority to ensure robustness and keep our system light-weight and easy-to-use. To improve accuracy and efficiency, we present several novelties to deal with DoFs setting, hand shape adjustment, self-interaction, tracking corresponding computation, and pose searching space constriction. These contributions make our system take the essence of wearable-based methods and model-based approaches, i.e., robustness and accuracy, but overcomes their downsides, i.e., tedious calibration, occlusions, and high dependency on GPU. Experimental results demonstrate that our system performs better than the state-of-the-art approach in the aspect of accuracy, robustness, and efficiency.

There are some factors in our system that should be considered in the future. When we compare our system with Remelli et al. [39] and Tkach et al. [40], we find that our system does not perform well on detailed hand shape tracking. We ascribe this weakness to the over constraints on the hand shape with the top 10 principal components of PCA as our shape parameters. More detailed and efficient shape parameters need to be designed for our triangular mesh model in the future. Besides, we do not claim to have “solved” the occlusion problem completely. When the occlusion occurs, we use the hand pose from the data glove to prevent tracking failure. The hand pose from our simple data glove is only approximate, and we expect an adoptive auto-calibration for the simple data glove by online learning in the future.

## Figures and Tables

**Figure 1 sensors-19-04680-f001:**
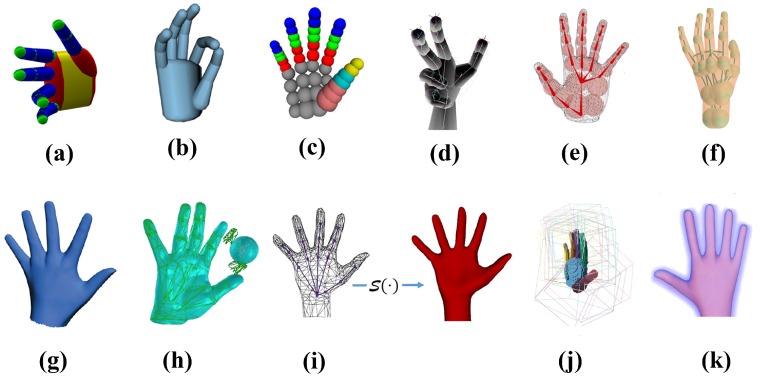
Hand models. (**a**) Capsule model [31,32,33]. (**b**) Cylinder model [27]. (**c**) Sphere model [23]. (**d**) Convex bodies for tracking [34]. (**e**) Sum of an-isotropic Gaussian model [35]. (**f**) Sphere-mesh model [5]. (**g**) Triangular hand model [4]. (**h**) Triangular mesh [8]. (**i**) Loop subdivision Surface of a triangular control mesh [36]. (**j**) Articulated Signed Distance Function (SDF) for a voxelized shape-primitive hand model [37]. (**k**) Articulated signed distance function for a hand model [1]. Images reproduced from the cited papers or their supplementary videos.

**Figure 2 sensors-19-04680-f002:**
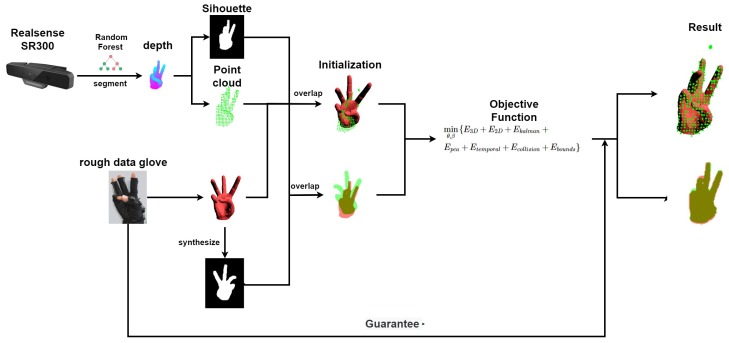
Overview of our system. Before tracking, the hand model has been prepared. During tracking, the workflow of the system is as follows: firstly, we acquire and process the inputs from the camera and the glove. For each acquired image, we extract a 3D point cloud and a 2D silhouette from the depth to provide the absolute measure. For the glove input, we get a rough hand pose for initialization. Secondly, we construct the objective function to measure the discrepancy between the hand model and the extracted 3D point cloud and 2D silhouette. Finally, we iteratively optimize the objective function by Levenberg–Marquardt approach to get the recovered hand motion. If the optimization failed, the input of the data glove provides a guarantee to ensure robustness.

**Figure 3 sensors-19-04680-f003:**
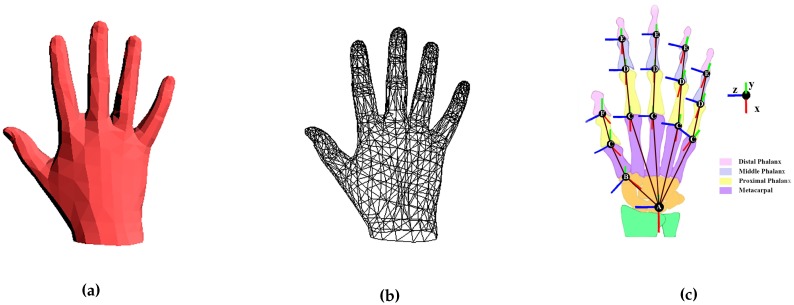
The MANO hand model and the kinematic chain. (**a**) The MANO hand model. (**b**) The triangular mesh of MANO. (**c**) The kinematic chain and human hand anatomy. The explanations of capital letters in (**c**) : **A** represents the Wrist joint. **B** represents the Carpometacarpal (CMC) joint. **C** represents the Metacarpophalangeal (MCP) joints. **D** represents the Proximal Interphalangeal (PIP) joints. **E** represents the Distal Interphalangeal (DIP) joints. **F** represents the Interphalangeal (IP) joint.

**Figure 4 sensors-19-04680-f004:**
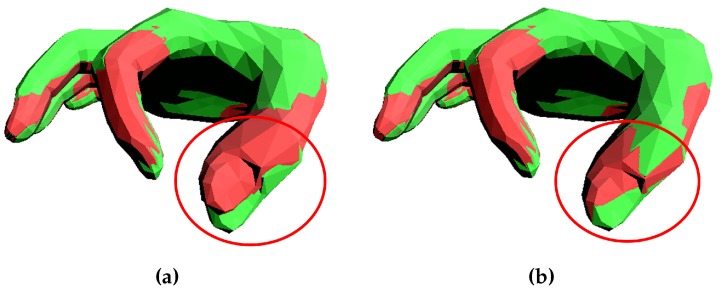
Fitting MANO hand model to the data set in [44]. Green one represents the data set, while red one is the fitted MANO model. (**a**) The IP joint with only one degree of freedom (DoF). (**b**) The IP joint with three DoFs.

**Figure 5 sensors-19-04680-f005:**
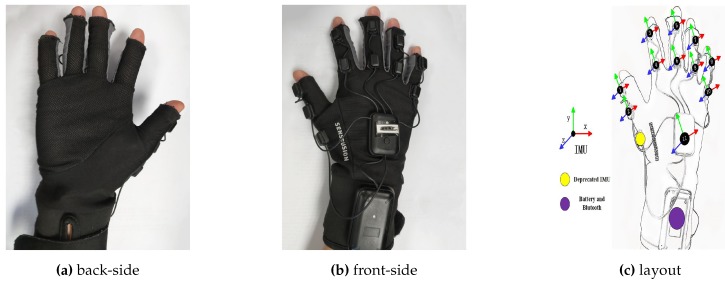
Data glove. (**a**) The back side of the data glove. (**b**) The front side of the data glove. (**c**) The location of inertial measurement units (IMUs) on the hand. We attach 11 IMUs on the glove made by breathable textile lining. Two IMUs on the distal, medial phalange of thumb; two IMUs on the medial, proximal phalanges of each of the other fingers; one IMU on the back of the hand.

**Figure 6 sensors-19-04680-f006:**
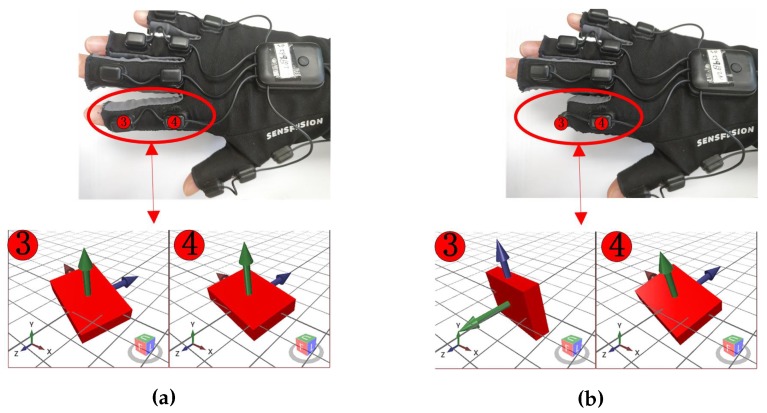
The example of how the IMUs attached on the index finger measure the rotation. (**a**) The initial poses of the 3rd and 4th IMU are equal $Q_{3}=Q_{4}$, when Index finger is stretching. (**b**) The relative pose $Q_{3,4}=Q_{3}^{*}*Q_{4}^{-1}$ of the 3rd and 4th IMU, when the index finger is blend. $Q_{3,4}$ can represent the 3rd IMU rotate 90 degrade counterclockwise around axis x related to the 4th IMU.

**Figure 7 sensors-19-04680-f007:**
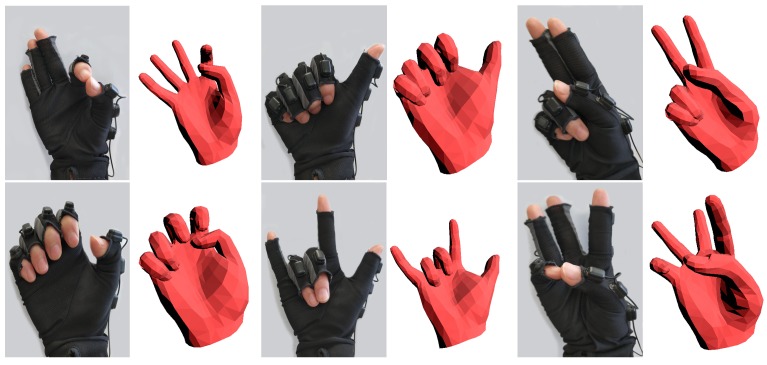
Some hand poses given by the simple data glove.

**Figure 8 sensors-19-04680-f008:**
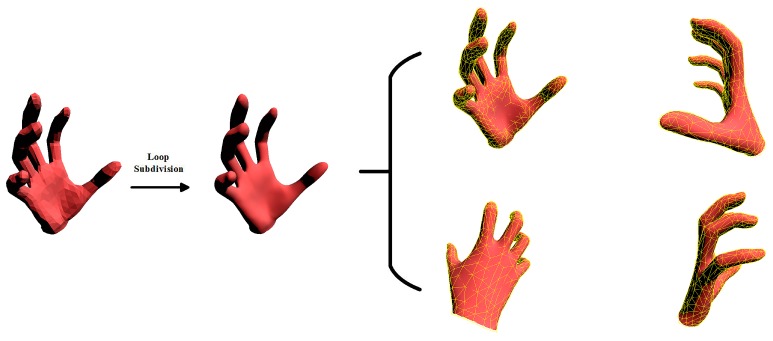
The original MANO hand model, the hand model applied the Loop subdivision, and four different views of the subdivided hand model with original MANO in wire-frame.

**Figure 9 sensors-19-04680-f009:**
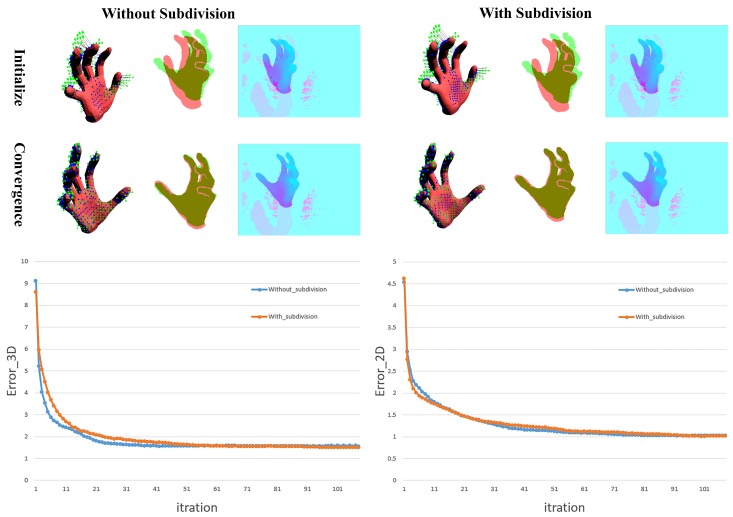
The comparison of two different corresponding points finding ways when we fit an input image from the Handy/Teaser data-set [5]. Each part from left to right is the 3D image, mixed 2D silhouette, and mixed depth image applied pseudo-color enhancement. The 3D image contains point cloud $X_{s}$ in green, hand model in red, and corresponding points on the hand model in blue. The red part in the mixed 2D silhouette is the rendered silhouette of the hand model, while the green part is the silhouette of input. The mixed depth shows how the rendered depth matches the original depth map, using pseudo-color enhancement for better viewing. The $Error_{3D}$ and $Error_{2D}$ are the metrics mentioned in Section 4.

**Figure 10 sensors-19-04680-f010:**
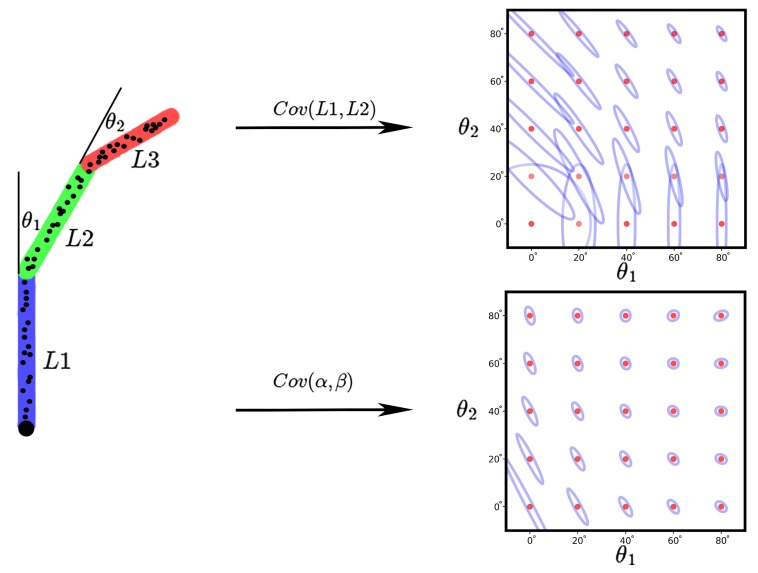
A visualization of the co-variance estimate for simple stick model as we vary the blend angles $\left\{\theta_{1},\theta_{2}\right\}$. the co-variance ellipsoids are centered at the corresponding $\left\{\theta_{1},\theta_{2}\right\}$ location. The explicit shape parameters $\left\{L1,L2,L3\right\}$ are the length of each stick, and the implicit shape parameters $\left\{\alpha ,\beta \right\}$ are the top two principal components analysis (PCA) components extract on 1000 random generated data.

**Figure 11 sensors-19-04680-f011:**
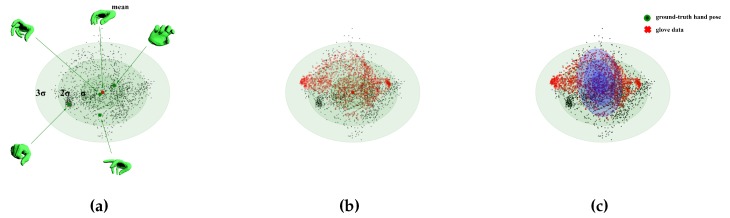
An illustration of the Gaussian distribution in the two-dimensional subspace of PCA. (**a**) The result $\tilde{\theta }$ of ground-truth hand pose(except the six global components) θ on the database of [44]. (**b**) The projection ${\tilde{\theta }}_{glove}$ of the glove data $\theta_{glove}$ into the same subspace in red dots. (**c**) The multivariate Gaussian model in blue color $N(\mu_{diff},\Sigma_{diff}^{\prime })$ for the difference between the glove data ${\tilde{\theta }}_{glove}$ and corresponding ground truth $\tilde{\theta }$ in the two-dimensional subspace.

**Figure 12 sensors-19-04680-f012:**
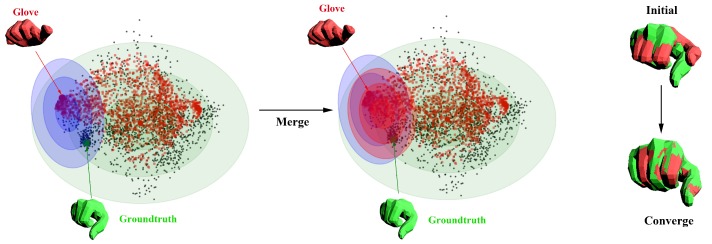
An illustration of the influence of N(0,Σ) and $N(\mu_{diff},\Sigma_{diff}^{\prime })$ and their fusion on a given glove input $\theta_{glove}^{i}$. The red hand model is the initialization from our data glove, while the green one is the ground-truth from the data-set [44]. The distribution of $N(\mu_{diff},\Sigma_{diff}^{\prime })$ is convert to $N({\tilde{\theta }}_{glove}^{i}-\mu_{diff},\Sigma_{diff}^{\prime })$ with the specific input of glove $\theta_{glove}^{i}$. The red ellipse represents the distribution of $N(\mu_{merge},\Sigma_{merge})$, which shows a smaller search area and a closer $\mu_{merge}$ toward the ground-truth. We also show the comparison between the initialization and convergence with the ground-truth on the right side.

**Figure 13 sensors-19-04680-f013:**
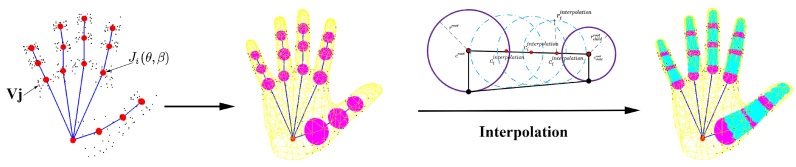
The process of generating a set of collision spheres to approximate the volume of the fingers in MANO hand model.

**Figure 14 sensors-19-04680-f014:**
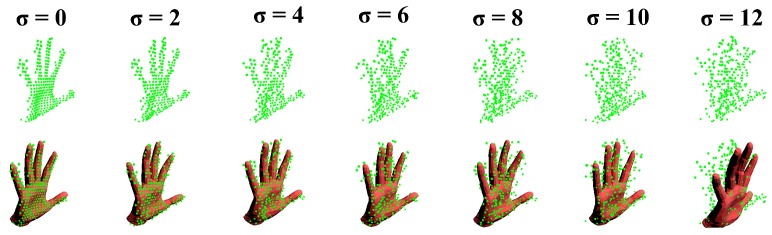
The influence of the standard deviation of noise on the point cloud from synthetic depth images. The visible performance of our system with those noise.

**Figure 15 sensors-19-04680-f015:**
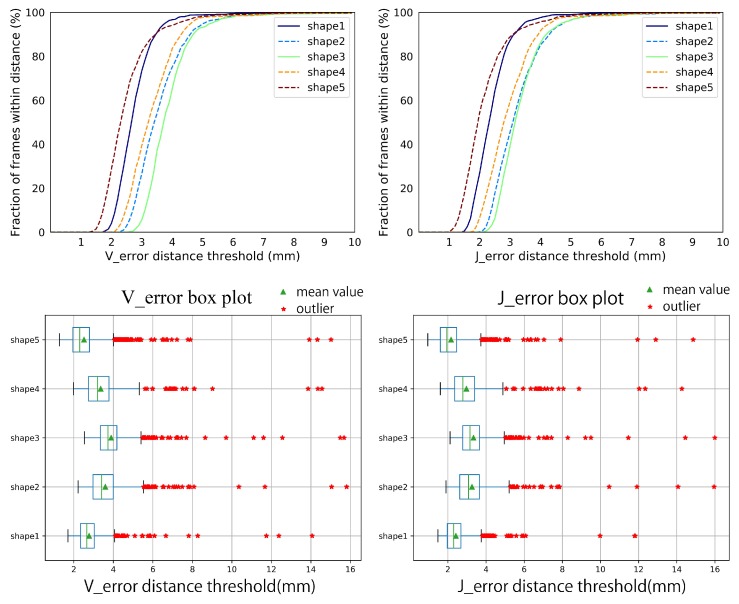
The $V_{error}$ and $J_{error}$ of our system on the five hand shapes and various hand poses in the synthetic data-set.

**Figure 16 sensors-19-04680-f016:**
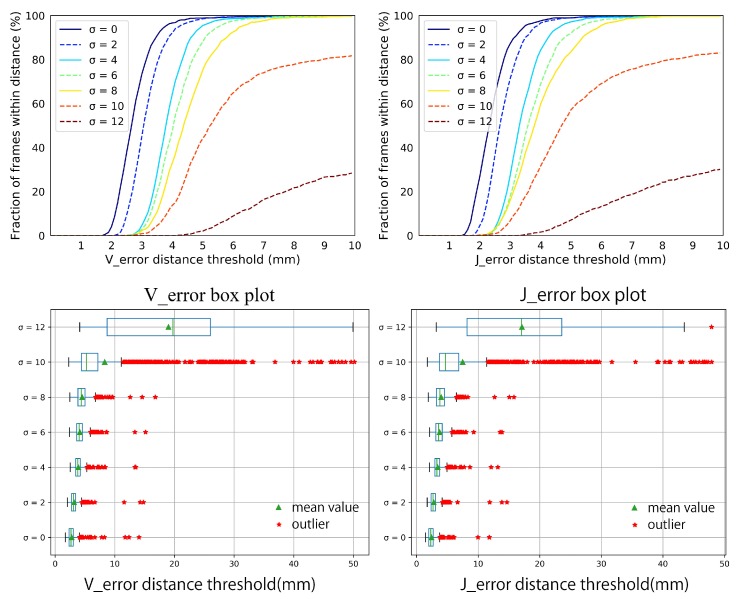
The $V_{error}$ and $J_{error}$ of our system with different noise in the synthetic data-set.

**Figure 17 sensors-19-04680-f017:**
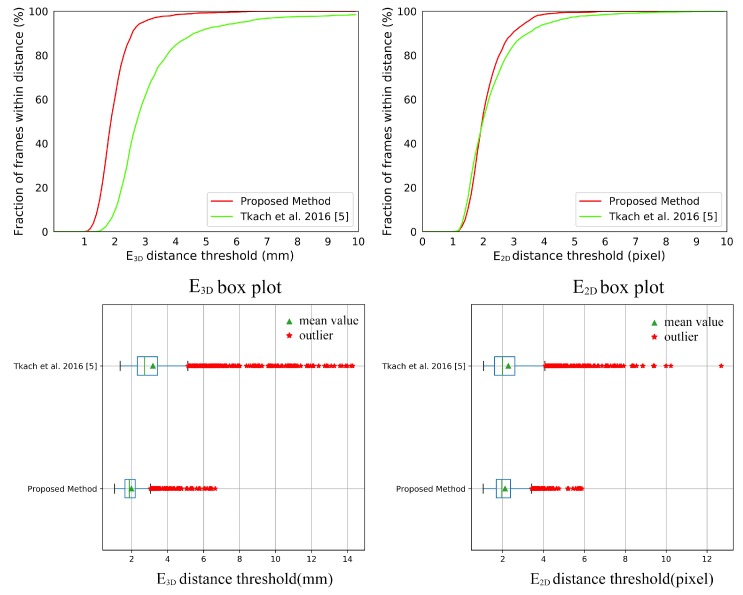
The $E_{3D}$ and $E_{2D}$ results of comparison between our system with [5] on Handy/ Teaser sequence.

**Figure 18 sensors-19-04680-f018:**
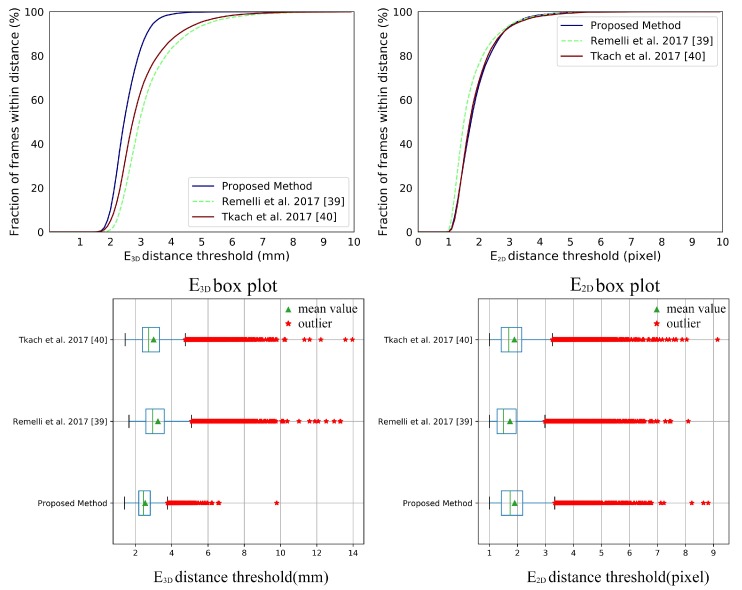
The $E_{3D}$ and $E_{2D}$ total results of comparison between our system with Remelli et al. [39] and Tkach et al. [40] on Handy/GuessWho sequence.

**Figure 19 sensors-19-04680-f019:**
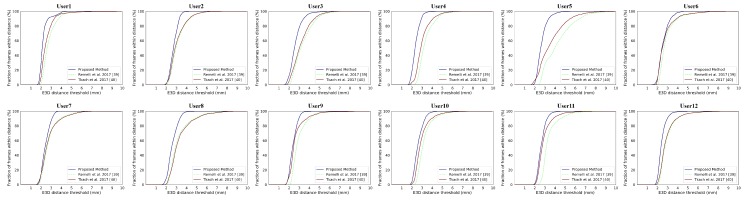
The $E_{3D}$ result of comparison between our system with Remelli et al. [39] and Tkach et al. [40] on the 12 subjects in Handy/GuessWho sequence.

**Figure 20 sensors-19-04680-f020:**
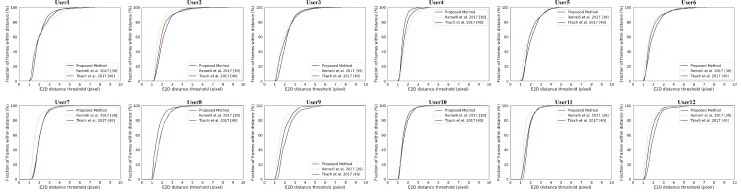
The $E_{2D}$ result of comparison between our system with Remelli et al. [39] and Tkach et al. [40] on the 12 subjects in Handy/GuessWho sequence.

**Figure 21 sensors-19-04680-f021:**
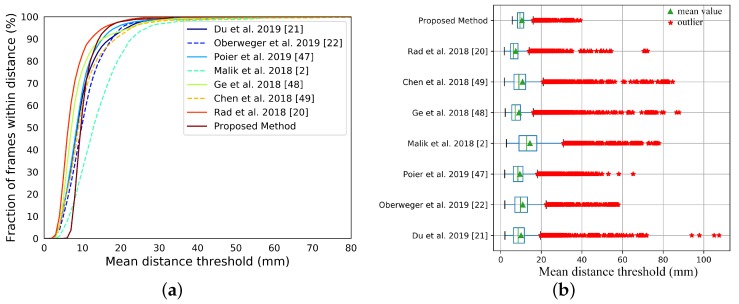
Experiment results on the NYU data-set. (**a**) The joints errors on the NYU data-set. (**b**) The box plot of the joints errors on the NYU data-set.

**Figure 22 sensors-19-04680-f022:**
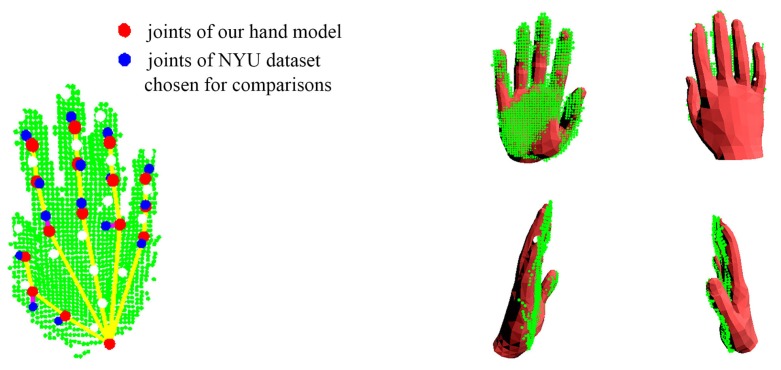
The subset of 15 joints chosen from the NYU annotation and the joints of our hand model in a simple hand pose. In this pose, the mean error of joints is 8.30 mm.

**Figure 23 sensors-19-04680-f023:**
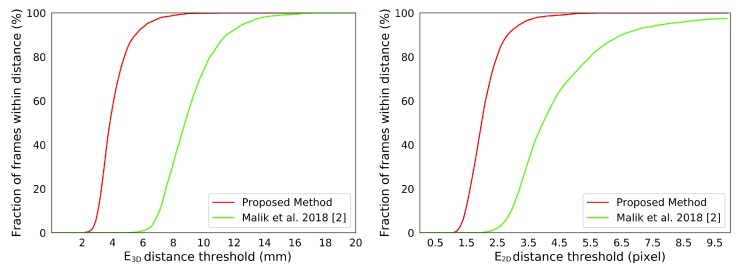
The $E_{3D}$ and $E_{2D}$ comparisons between our system with Malik et al. [2] on the NYU data-set.

**Figure 24 sensors-19-04680-f024:**
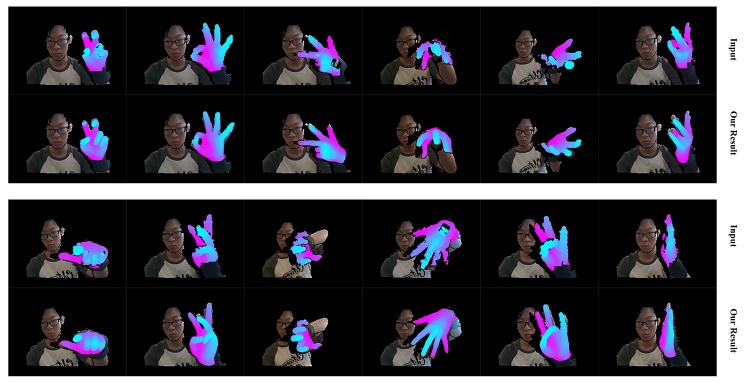
The real-time performance of our system. There are two rows. For each row, the upper one shows the hand segment of the input depth after pseudo-color enhancement, while the lower one shows the rendered depth image from the recovered hand model after pseudo-color enhancement.

**Figure 25 sensors-19-04680-f025:**
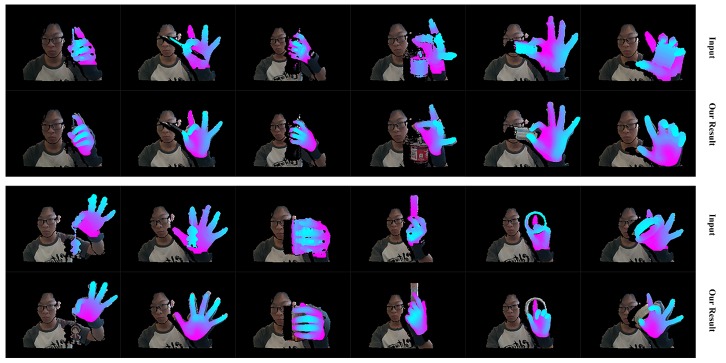
The real-time performance of our system with occlusion from some small items. There are two rows. For each row, the upper one shows the hand segment of the input depth after pseudo-color enhancement, while the lower one shows the rendered depth image from the recovered hand model after pseudo-color enhancement.

**Figure 26 sensors-19-04680-f026:**
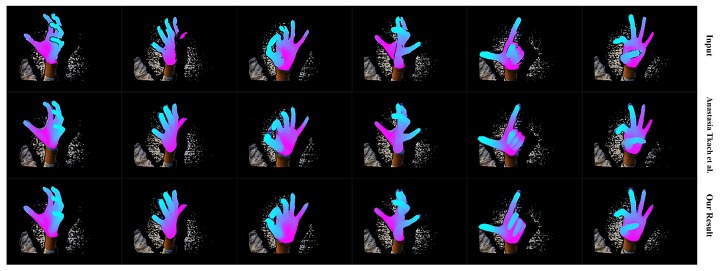
The qualitative comparison between our system with Tkach et al. [5] on Handy/Teaser. The upper row shows the hand segment of the input depth after pseudo-color enhancement; the middle row shows the rendered depth image from the recovered hand model of Tkach et al. [5]; the bottom row shows the rendered depth image from the recovered hand model of our system after pseudo-color enhancement.

**Figure 27 sensors-19-04680-f027:**
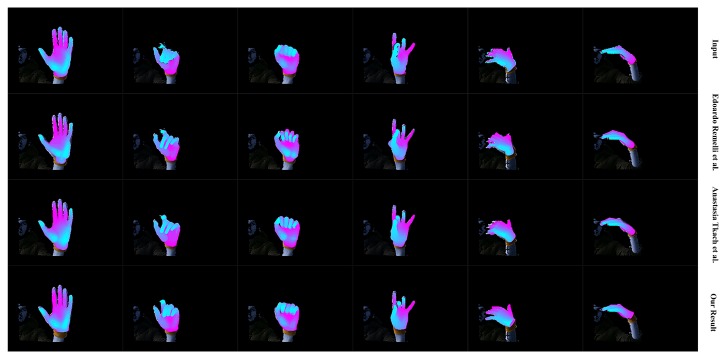
The qualitative comparison between our system with Remelli et al. [39] and Tkach et al. [40] on the Handy/GuessWho sequence of user4. The first row shows the hand segment of the input depth after pseudo-color enhancement; the second row shows the rendered depth image from the recovered hand model of Remelli et al. [39]; the third row shows the rendered depth image from the recovered hand model of Tkach et al. [40]; the bottom row shows the rendered depth image from the recovered hand model of our system.

**Figure 28 sensors-19-04680-f028:**
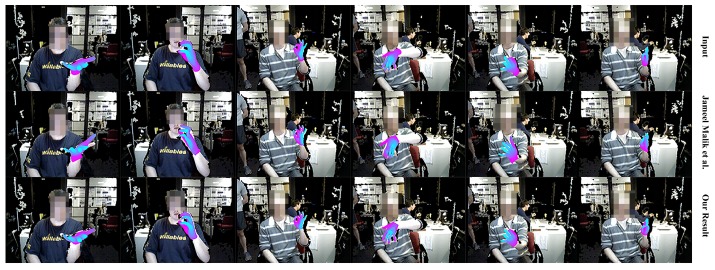
The qualitative comparison between our system with Malik et al. [2] on the NYU data-set. The first row shows the hand segment of the input depth after pseudo-color enhancement; the second row shows the rendered depth image from the recovered hand model of Malik et al. [2]; the bottom row shows the rendered depth image from the recovered hand model of our system.

**Table 1 sensors-19-04680-t001:** The extracted limitations of pose parameters. We extract those limitations by fitting MANO model to the database provided by [44] using our kinematic structure.

**Thumb**	**CMC_x**	**CMC_y**	**CMC_z**	**DCP_x**	**DCP_y**	**DCP_z**	**IP_x**	**IP_y**	**IP_z**
Max	18.37°	29.34°	78.16°	16.36°	33.72°	21.28°	17.59°	78.29°	19.26°
Min	3.08°	−33.06°	−24.01°	−18.85°	−75.38°	−19.70°	−13.11°	−93.88°	−15.56°
**Index**	**DCP_x**	**DCP_y**	**DCP_z**	**PIP_x**	**PIP_y**	**PIP_z**	**DIP_x**	**DIP_y**	**DIP_z**
Max	18.53°	22.72°	89.40°	16.74°	7.66°	101.89°	11.71°	5.96°	75.80°
Min	−15.41°	−31.75°	89.40°	−11.08°	−4.19°	−34.83°	−13.58°	−4.43°	−42.08°
**Middle**	**DCP_x**	**DCP_y**	**DCP_z**	**PIP_x**	**PIP_y**	**PIP_z**	**DIP_x**	**DIP_y**	**DIP_z**
Max	16.65°	18.74°	105.90°	14.43°	4.17°	101.65°	6.98°	6.27°	77.22°
Min	−13.19°	−23.33°	−39.66°	−13.86°	−6.03°	−30.38°	−7.28°	−1.24°	−28.96°
**Ring**	**DCP_x**	**DCP_y**	**DCP_z**	**PIP_x**	**PIP_y**	**PIP_z**	**DIP_x**	**DIP_y**	**DIP_z**
Max	17.53°	46.82°	106.52°	16.60°	6.69°	101.61°	7.27°	6.28°	82.49°
Min	−20.31°	−30.92°	−60.50°	−19.58°	−5.70°	−24.83°	−15.63°	−5.45°	−30.86°
**Pinky**	**DCP_x**	**DCP_y**	**DCP_z**	**PIP_x**	**PIP_y**	**PIP_z**	**DIP_x**	**DIP_y**	**DIP_z**
Max	18.47°	31.52°	102.32°	2.48°	7.11°	103.65°	10.22°	6.63°	82.53°
Min	−16.75°	−19.83°	−39.75°	−16.44°	−2.81°	−29.03°	−11.20°	−0.12°	−32.14°

**Table 2 sensors-19-04680-t002:** The numerical improvements between our system and [5] on the Handy/Teaser sequence. We consider the the mean value and the standard deviation (SD) of the $E_{3D}$ and $E_{2D}$.

		[5]	Proposed Method	Improvement on [5]
$E_{3D}$	Mean (mm)	3.18	1.99	37%
	SD (mm)	1.61	0.61	62%
$E_{2D}$	Mean (pixel)	2.27	2.12	6%
	SD (pixel)	1.06	0.64	40%

**Table 3 sensors-19-04680-t003:** The results of significance tests on the $E_{3D}$ and $E_{2D}$ for the Handy data-set.

	Handy/Teaser on [5]		Handy/GuessWho on [39]		Handy/GuessWho on [40]	
	$E_{3D}$	$E_{2D}$	$E_{3D}$	$E_{2D}$	$E_{3D}$	$E_{2D}$
*p*-value	$1.53\times e^{-262}$	$6.57\times e^{-12}$	0	$8.43\times e^{-156}$	0	0.1907

**Table 4 sensors-19-04680-t004:** The numerical improvements between our system and [39,40] on the Handy/GuessWho sequence. We consider the the mean value and the SD of the $E_{3D}$ and $E_{2D}$.

		[39]	[40]	Proposed Method	Improvement on [39]	Inprovement on [40]
$E_{3D}$	Mean (mm)	3.24	3.00	2.54	22%	15%
	SD (mm)	1.02	0.97	0.50	50%	48%
$E_{2D}$	Mean (pixel)	1.73	1.89	1.90	−9%	−0.5%
	SD (pixel)	0.71	0.71	0.68	4%	4%

**Table 5 sensors-19-04680-t005:** The numerical improvements between our system with [2,20,21,22,47,48,49] on the NYU data-set. We consider the the mean value and the standard deviation(SD) of the joints error.

	Ours	[2]	[20]	[21]	[22]	[47]	[48]	[49]
Mean (mm)	10.45	14.42	7.44	10.08	10.89	9.47	9.05	10.78
SD (mm)	3.64	8.30	4.44	6.67	6.22	5.27	7.02	8.00
Mean Improvement	–	27%	−40%	−3%	4%	−10%	−15%	3%
SD Improvement	–	56%	18%	45%	41%	30%	48%	54%

**Table 6 sensors-19-04680-t006:** The results of significance tests between our system with [2,20,21,22,47,48,49] on the joints error of the NYU data-set.

	With [2]	With [20]	With [21]	With [22]	With [47]	With [48]	With [49]
*p*-value	0	0	$1.07\times e^{-5}$	$1.74\times e^{-8}$	$1.13\times e^{-43}$	$7.39\times e^{-58}$	0.0007

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
