# Peer review of "A Model-Based System for Real-Time Articulated Hand Tracking Using a Simple Data Glove and a Depth Camera"

_sensors, 2019, doi:10.3390/s19214680_

Round 1
Reviewer 1 Report
This paper describes a model-based system for hand tracking in real time using a Depth Camera and a Data Glove. The paper describes an interesting system that includes several technologies. The main contribution is the combination of these technologies. From my point of view, there several aspects that must be improved before publication:
I am not an English person but I think there are several errors that must be corrected: line 220 “, These” line 384 don’t, (not contractions) line 386 “In Figure 22 show…?” line 389 Finger 23, 24, 25. Figures??. I suggest reviewing the paper by an English person. Also revise the verb tense: line 314 and 321: is -> was I’d propose including a glossary with all the abbreviations and initials. (ICP, PCA,…) Figure 9. The graphs are very small. I suggest dividing the figure in two Section 4.2. It is necessary to justify more in detail WHY do you use different metrics for different datasets? Caption in figure 15. “on different noise” -> “with different noise”? Lines 350-360. Discuss the influence of the hand model in the results. Figures 18-20: The font is very small. It is necessary to increase the font In order to validate that the improvement is statistically significant, it is necessary to do a significance test. Significance tests must be included in the paper. Finally, the authors claim an efficiency improvement but there is not a comparison with previous works on time consuming, for example. Efficiency results must be included in order to claim this contribution.Author Response
Response to Reviewer 1 Comments
We are truly grateful for your critical comments and thoughtful suggestions. Based on these comments and suggestions, we have made careful modifications to the original manuscript. All changes made to the text are in red color. Below you will find our point-by-point responses to the comments/ questions. The original comments are in black, and our responses are in red. (better view the pdf)
Point 1: I think there are several errors that must be corrected: line 220 “, These” line 384 don’t, (not contractions) line 386 “In Figure 22 show…?” line 389 Finger 23, 24, 25. Figures??. I suggest reviewing the paper by an English person. Also revise the verb tense: line 314 and 321: is -> was
Response 1: We regret there were problems with the English. We have corrected these errors. The corrected results can be found in
line 228 for “line 220 “, These””,
line 411 for “line 384 don’t” ,
line 412 for line 386 “In Figure 22 show…?”,
line 416 for “line 389 Finger 23, 24, 25. Figures??” ,
line 322 and line 329 for the verb tense of “line 314 and 321: is -> was”.
Besides, we have checked this paper again to prevent these English errors.
Point 2: I’d propose including a glossary with all the abbreviations and initials. (ICP, PCA,…)
Response 2: Thanks for your advice. We have included an “Abbreviations” Section at the end of the article. In the “Abbreviations” Section, we include all the abbreviations (e.g. HCI, DoF, ICP, PCA, PSO).
Point 3: Figure 9. The graphs are very small. I suggest dividing the figure in two
Response 3: Yes, we have divided Figure 9 in two.
Point 4: Section 4.2. It is necessary to justify more in detail WHY do you use different metrics for different datasets?
Response 4: Thanks for point out this issue. The metrics are used to explain the difference between the recovered hand motion with the original hand motion. Different data-sets offer the original hand motion in different ways. That is why different metrics are used.
We have added the detailed reasons for choosing metrics for different datasets in the revised manuscript. You can find this new content in Section 4.2 in red color in our revised manuscript.
Point 5: Caption in figure 15. “on different noise” -> “with different noise”?
Response 5: Yes, we have corrected this error. In our new manuscript, Figure 15 changes into Figure 16, you can find the correction in Figure 16 in red.
Point 6: Lines 350-360. Discuss the influence of the hand model in the results.
Response 6: The influence of the hand model reflected in how well it fits the surface geometry of the human hand under different poses and the whole hand shape. We have added a discussion for the performance of the hand model. You can find this in line 375-379 in our revised manuscript.
Point 7: Figures 18-20: The font is very small. It is necessary to increase the font
Response 7: We have increased the front.
Point 8: In order to validate that the improvement is statistically significant, it is necessary to do a significance test. Significance tests must be included in the paper.
Response 8: Thanks for your suggestion. We have conducted significant tests between our results with each of the works we compared. The p-values of significant tests are added in the article in Table 4 and Table 6. Related discussion is also added in the revised manuscript.
Point 9: Finally, the authors claim an efficiency improvement but there is not a comparison with previous works on time-consuming, for example. Efficiency results must be included in order to claim this contribution.
Response 9: We claim an efficiency improvement by considering the dependency on GPU. GPU is always used to accelerate heavy computational tasks by high parallelization. We have added a new Section “System efficiency” that gives the runtime hardware cost and the performance of our system to show the efficiency improvement.

Reviewer 2 Report
In general, the methodology, results and discussion are well presented. This work is publishable upon addressing the following comments.
Abstract, please write “Full name (Abbreviation)” instead of “Abbreviation (Full name)”. For instance “HCI(human-computer interaction)”, and “DoFs(degree of freedom)” should be revised. Abstract, authors are suggested to highlight the performance of proposed work numerically, along with the improvement by proposed work. Keywords, more representative and relevant keywords should be included. For instance, depth camera should be one of the keywords. Introduction, paragraph 1, line 20, please define the abbreviations “VR/AR”. Introduction, authors cite too many references which suppose that literature review will be covered in section 2. Please shorten the coverage in section 1. Figure 1 should not be discussed in introduction. It is the proposed work. Similarly, please revise “IMUs(Inertial Measurement Unit)” in line 53. Please state clearly the contributions of this paper, preferably in point-form. Please ensure that proper spacing is given. Some wordings/symbols are linked together without spacing. Section 2, please update the references in general by including more coverage on recently published articles (2015-2019). Section 2, it is not recommended to use present tense. Section 2, authors briefly introduced various existing works. However, it is expected to have a summary on their inadequacies and limitations, which is followed by the rationale of the proposed work. It can be seen that the methodology is well written in a way that authors explain every step with the aid of figures. It would be great if authors could indicate the steps (step 1, step 2, step 3,…) in Figure 1 and recall the step number in written description. Figure 11, legends should be given. Authors are suggested to highlight the percentage improvement by proposed work compared to related works. Section 4.4, why it is titled qualitative results?
Author Response
Response to Reviewer 2 Comments
We are truly grateful for your critical comments and thoughtful suggestions. Based on these comments and suggestions, we have made careful modifications to the original manuscript. All changes made to the text are in red color. Below you will find our point-by-point responses to the comments/ questions. The original comments are in black, and our responses are in red. (better view the pdf)
Point 1: Abstract, please write “Full name (Abbreviation)” instead of “Abbreviation (Full name)”. For instance “HCI(human-computer interaction)”, and “DoFs(degree of freedom)” should be revised.
Similarly, please revise “IMUs(Inertial Measurement Unit)” in line 53.
Response 1: Thanks for pointing out this problem. We have corrected these errors. The revised results can be found in line 2, line 9 in our revised manuscript. And we have included an “Abbreviations” Section at the end of the article. In the “Abbreviations” Section, we explain all the abbreviations (e.g. HCI, DoF, ICP, PCA, PSO).
Point 2: Abstract, authors are suggested to highlight the performance of proposed work numerically, along with the improvement by proposed work.
Response 2: We are sorry for not highlight the performance of our work numerically. We have added this part in the abstract, from line 13 to line 16, in our revised manuscript.
Point 3: Keywords, more representative and relevant keywords should be included. For instance, depth camera should be one of the keywords.
Response 3: It is really true as suggested that we should include “depth camera” as one of the keywords. We have included “depth camera” as one of the keywords in our revised manuscript.
Point 4: Introduction, paragraph 1, line 20, please define the abbreviations “VR/AR”.
Response 4: We have added the definition for the abbreviations “VR/AR”, line 21. “VR” means virtual reality and “AR” is augmented reality. We also have included the two abbreviations in the Abbreviations Section.
Point 5: Introduction, authors cite too many references which suppose that literature review will be covered in section 2. Please shorten the coverage in section 1.
Response 5: According to your suggestion, we have chosen some representative references to shorten the coverage in Section 1.
Point 6: Figure 1 should not be discussed in introduction. It is the proposed work.
Response 6: Thanks for pointing out this problem. We have removed the discussion in introduction. And Figure 1 has been moved to the Method Section (Section 3).
Point 7: Please state clearly the contributions of this paper, preferably in point-form.
Response 7: Thanks for your suggestion. We have re-organized the statement of our contributions in point-form. There are four contributions in the following four aspects:
1)The function of our simple data glove; 2)The DoFs setting; 3)The simplified fitting terms; 4)The strategy for the shape adjustment of the triangular mesh model.
The detailed revised result can be found from line 53 to line 68 in our revised manuscript.
Point 8: Please ensure that proper spacing is given. Some wordings/symbols are linked together without spacing.
Response 8: We are very sorry for our negligence of proper spacing. We have corrected these errors.
Point 9: Section 2, please update the references in general by including more coverage on recently published articles (2015-2019).
Response 9: Yes, we have removed or updated most of the references published before 2015. Some representative references in the early years are kept.
Point 10: Section 2, it is not recommended to use present tense.
Response 10: Thanks for pointing out this problem. We have corrected the tense in Section 2. In our revised manuscript, you can find these revised results in red in Section 2.
Point 11: Section 2, authors briefly introduced various existing works. However, it is expected to have a summary on their inadequacies and limitations, which is followed by the rationale of the proposed work.
Response 11: Thanks for your advice. We have added the summary on inadequacies and limitations followed by the rationale of our work. These revised results can be found in red color in Section 2.
Point 12: It can be seen that the methodology is well written in a way that authors explain every step with the aid of figures. It would be great if authors could indicate the steps (step 1, step 2, step 3,…) in Figure 1 and recall the step number in written description.
Response 12: According to your suggestion, Figure 1 has been moved to the Method Section (Section 3). And we have indicated the workflow of our system step by step in the caption of original Finger 1 and recalled these steps in the overview of Section 3 from line 204 to line 209.
Point 13: Figure 11, legends should be given.
Response 13: Yes, we have added the legends for the green point and red point in Finger 11.
Point 14: Authors are suggested to highlight the percentage improvement by proposed work compared to related works
Response 14: Thanks for your advice. We have added Table 2, Table 3, and Table 5 to show the percentage improvement on accuracy and robustness.
Point 15: Section 4.4, why it is titled qualitative results?
Response 15: The reason why we titled qualitative results for Section 4.4 is that we want to show the qualitative performance of our system in Section 4.4. Maybe the title is a little confusing; we have re-named it as “Qualitative experiments”.

Round 2
Reviewer 1 Report
The authors have addressed properly my comments so it can be accepted.
Author Response
Our deepest gratitude goes to the anonymous reviewer for these careful and thoughtful suggestions that have helped improve this paper substantially.
Reviewer 2 Report
I have some follow-up comments:
Point 7: Please state clearly the contributions of this paper, preferably in point-form.
Response 7: Thanks for your suggestion. We have re-organized the statement of our contributions in point-form. There are four contributions in the following four aspects:
1)The function of our simple data glove; 2)The DoFs setting; 3)The simplified fitting terms; 4)The strategy for the shape adjustment of the triangular mesh model.
The detailed revised result can be found from line 53 to line 68 in our revised manuscript.
Follow-up comment: One more point could be added: The proposed work improves the performance by xx% compared to existing works.
Point 8: Please ensure that proper spacing is given. Some wordings/symbols are linked together without spacing.
Response 8: We are very sorry for our negligence of proper spacing. We have corrected these errors.
Follow-up comment: There are still various places that have linked wordings/symbols. For instance, line 26: “freedoms(DoFs)”, line 76: “Laura Dipietro et al.[11]”, line 84: “Jameel Malik et al.[2]”, line 156: “Most works[5,7,23,27,28,30,32,36]”, line 171: “Chen Qian et al.[23]”, line 182: “deformations[30,36],”, line 286: “Anastasia Tkach et al.[40]”.
Author Response
Response to Reviewer 2 Follow-up Comments
We are grateful for the time and effort that you have put into reviewing the previous version of the manuscript. Based on the instructions provided in your comments, we uploaded the file of the revised manuscript. All changes made to the text are in Blue color in our revised manuscript. Below you will find our point-by-point responses to the comments/ questions. The original comments are in black, and our responses are in red. (better view the pdf)
Follow-up comment 1 for Point 7: One more point could be added: The proposed work improves the performance by xx% compared to existing works.
Response 1: Thanks for the suggestion. We have add one more point that shows the improvements by xx% compared to existing works. It can be found in line 56-58.
Follow-up comment 2 for Point 8: There are still various places that have linked wordings/symbols. For instance, line 26: “freedoms(DoFs)”, line 76: “Laura Dipietro et al.[11]”, line 84: “Jameel Malik et al.[2]”, line 156: “Most works[5,7,23,27,28,30,32,36]”, line 171: “Chen Qian et al.[23]”, line 182: “deformations[30,36],”, line 286: “Anastasia Tkach et al.[40]”.
Response 2: We apologize for not understanding where the proper spacing should be given before and leaving so much similar errors. From the examples you gave, we have corrected all these errors in our revised manuscript. These corrections in blue color can be found in our revised manuscript. For finding these corrections fast, we list the lines and Figures as follows:
line 26, line 79, line 87, line 159, line 174, line 185, line 218, line 220, line 223, line 279, line 289, line 301, line 335, line 370, line 401, line 405, line 416–418, line 442, and Figure 1, Figure 18, Figure 19, Figure 20, Figure 23, Figure 26, Figure 27, Figure 28,
